# A FRAMEWORK FOR BENCHMARKING CLASS-OUT-OF-DISTRIBUTION DETECTION AND ITS APPLICATION TO IMAGENET

**Ido Galil\***
Technion
idogalil.ig@gmail.com

**Mohammed Dabbah\***
Amazon
m.m.dabbah@gmail.com

**Ran El-Yaniv**
Technion,   Deci.AI
rani@cs.technion.ac.il

## ABSTRACT

When deployed for risk-sensitive tasks, deep neural networks must be able to detect instances with labels from outside the distribution for which they were trained. In this paper we present a novel framework to benchmark the ability of image classifiers to detect class-out-of-distribution instances (i.e., instances whose true labels do not appear in the training distribution) at various levels of detection difficulty. We apply this technique to ImageNet, and benchmark 525 pretrained, publicly available, ImageNet-1k classifiers. The code for generating a benchmark for any ImageNet-1k classifier, along with the benchmarks prepared for the above-mentioned 525 models is available at https://github.com/mdabbah/COOD_benchmarking.

The usefulness of the proposed framework and its advantage over alternative existing benchmarks is demonstrated by analyzing the results obtained for these models, which reveals numerous novel observations including: (1) knowledge distillation consistently improves *class-out-of-distribution* (C-OOD) detection performance; (2) a subset of ViTs performs better C-OOD detection than any other model; (3) the language—vision CLIP model achieves good zero-shot detection performance, with its best instance outperforming 96% of all other models evaluated; (4) accuracy and in-distribution ranking are positively correlated to C-OOD detection; and (5) we compare various confidence functions for C-OOD detection. Our companion paper, also published in ICLR 2023 (Galil et al., 2023), examines the uncertainty estimation performance (ranking, calibration, and selective prediction performance) of these classifiers in an in-distribution setting.

## 1 INTRODUCTION

Deep neural networks (DNNs) show great performance in a wide variety of application domains including computer vision, natural language understanding and audio processing. These models are trained on data coming from a certain distribution $P(X, Y)$, usually with the assumption that test points will be sampled from the same distribution. When the underlying distribution $P(X, Y)$ of test points is different from the one used to train a model, we may no longer expect the same performance from the model. The difference in distribution may be the result of many processes such as natural deviation in the input space $\mathcal{X}$, noisy sensor readings of inputs, abrupt changes due to random events, newly arrived or refined input classes, etc. Here we distinguish between input distributional changes in $P_{X|Y}$ and changes in the label distribution. We focus on the latter case and consider the *class-out-of-distribution* (C-OOD) scenario, AKA *open-set recognition* (Scheirer et al., 2013), where the label support set $\mathcal{Y}$ changes to a different set that includes the set $\mathcal{Y}_{\text{OOD}}$, containing new classes not observed in training.

Consider the *detection* task in which our model is required to distinguish between samples belonging to classes it has seen in training, where $x \sim P(x|y \in \mathcal{Y}_{\text{ID}})$, and samples belonging to novel classes, i.e., $x \sim P(x|y \in \mathcal{Y}_{\text{OOD}})$. The question we now ask is: how should models be evaluated to most accurately reflect their detection performance? We aim to benchmark the detection performance

---

*The first two authors have equal contribution.

of DNN classification models that use their *confidence rate function* $\kappa$ (e.g., softmax response; see Section 2) to detect OOD labels, where the basic premise is that instances whose labels are in $\mathcal{Y}_{OOD}$ are assigned lower $\kappa$ values.

Most works on OOD detection use small-scale datasets that generally do not resemble the training distribution and, therefore, are easy to detect. The use of such sets often causes C-OOD detectors to appear better than they truly are when faced with realistic, yet harder tasks. Motivated by this deficiency, Hendrycks et al. (2021) introduced the ImageNet-O dataset as a solution. ImageNet-O, however, has two limitations. First, it benchmarks models with a single difficulty level exclusively, having only hard C-OOD instances, which might not be relevant for every task's requirements (Section 3 explains how to define different difficulty levels). Second, the original intent in the creation of ImageNet-O was to include only hard C-OOD instances. Its definition of "OOD hardness", however, was carried out with respect to ResNet-50's difficulty in detecting C-OOD classes, specifically when using softmax as its confidence function. This property makes ImageNet-O strongly biased. Indeed, consider the right-most box in Figure 1, which corresponds to the performance of 525 models over ImageNet-O. The orange dot in that box corresponds to ResNet-50, whose OOD detection performance is severely harmed by these ImageNet-O data. Nevertheless, it is evident that numerous models perform quite well, and all other models perform better than ResNet-50. The lack of an objective benchmark for C-OOD is the main motivation for our work.

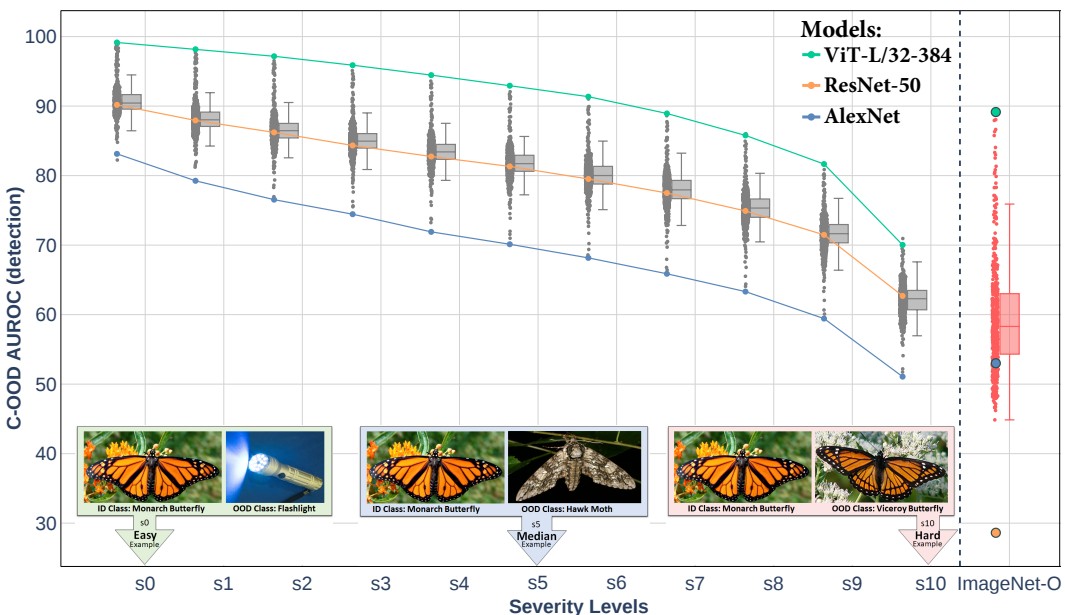

Figure 1: OOD performance across severity (difficulty) levels, using the benchmarks produced by our framework. The detection performance decreases for all models as we increase the difficulty until it reaches near chance detection performance at the highest severity ($s_{10}$). The top curve belongs to ViT-L/32-384, which surpasses all models at every severity level. We also observe how success or failure with regard to the previous C-OOD benchmark, ImageNet-O, does not reflect the models' true OOD detection performance since it was designed to specifically fool ResNet-50. At the bottom we provide visual examples for OOD classes from ImageNet-21k that may populate each severity level due to their similarity to ID classes from ImageNet-1k, and in this example, to a Monarch butterfly.

**Our contributions.** We propose a novel technique to generate a C-OOD benchmark that covers a variety of difficulty levels. Unlike other existing benchmarks (e.g., ImageNet-O), our technique is not biased towards an arbitrary model such as Resnet50 and/or a specific confidence function such as the softmax response. This useful property is obtained by tailoring the benchmark to the model being evaluated, including its confidence function, and not seeking to determine a single objective criterion for hardness of C-OOD samples (see Section 3).

Second, we show and explain how we filter ImageNet-21k to use it for the purpose of generating C-OOD benchmarks for ImageNet-1k (Deng et al., 2009) classifiers (see Section 4). We will provide

a simple code to choose the filtering parameters most suitable for the specific aim for which the benchmark is meant (e.g., what is classes are considered OOD).

Third, we demonstrate the power and usability of our method by applying our C-OOD framework to generate benchmarks for 525 ImageNet-1k classifiers available from popular repositories. We provide a benchmark for each of these classifiers, which will be available for use from our code.

We then analyze the results of these benchmarks to make numerous novel observations concerning C-OOD detection such as: (1) training regimes using knowledge distillation (Hinton et al., 2015) consistently yield models with better C-OOD detection performance than the same models trained identically, but without distillation; (2) a subset of ViTs performs better C-OOD detection than any other model; (3) the language–vision model CLIP achieves good zero-shot detection performance for low difficulty (severity) levels; (4) accuracy and in-distribution (ID) ranking are positively correlated with C-OOD detection; (5) we compare the performance of various confidence functions for C-OOD detection; (6) A number of other observations (see Section 5).

Lastly, we emphasize that the resulting difficulty levels of our framework allow benchmarking with respect to the difficulty levels most relevant to the task. For example, for a task with a high tolerance for risk (e.g., a task for an entertainment application), the performance of a model on a median difficulty level might be more important than on the hardest difficulty level (severity 10). The opposite might be true for some applications with a low tolerance for risk (e.g., medical applications), for which one requires the best performance to be attained even if the OOD is very hard to detect (severity 10). Furthermore, in Section 5 we show that detection algorithms do not always improve performance on all inputs equally, and could even hurt performance for specific difficulty levels and models (see Figure 7 for a striking example). Choosing the combination of (model, detection algorithm) based only on the detection performance on all data may yield sub-optimal results for our specific desired level of difficulty.

## 2 PROBLEM SETUP

Let $\mathcal{X}$ be the input space and $\mathcal{Y} = \mathcal{Y}_{\text{ID}} \cup \mathcal{Y}_{\text{OOD}}$ be the label space. Let $P(\mathcal{X}, \mathcal{Y})$ be an unknown distribution over $\mathcal{X} \times \mathcal{Y}$. A model $f$ is a prediction function $f : \mathcal{X} \rightarrow \mathcal{Y}_{\text{ID}}$, and its predicted label for an image $x$ is denoted by $\hat{y}_f(x)$. The model $f$ is produced by training on a labeled set $T_m = \{(x_i, y_i)\}_{i=1}^m \subseteq (\mathcal{X} \times \mathcal{Y}_{\text{ID}})$, sampled i.i.d. from $P(\mathcal{X}, \mathcal{Y}_{\text{ID}})$, with the objective of minimizing its *empirical risk*, defined by $\hat{r}(f|T_m) \triangleq \frac{1}{m} \sum_{i=1}^m \ell(f(x_i), y_i)$, where $\ell : \mathcal{Y}_{\text{ID}} \times \mathcal{Y}_{\text{ID}} \rightarrow \mathbb{R}^+$ is a given loss function (e.g., cross-entropy loss for classification). Note that by this definition, the model $f$ will always misclassify any $x \sim P(\mathcal{X}, \mathcal{Y}_{\text{OOD}})$.

We define a *confidence score* function $\kappa(x, \hat{y}|f)$, where $x \in \mathcal{X}$, and $\hat{y} \in \mathcal{Y}_{\text{ID}}$ is the model's prediction for $x$, as follows. The function $\kappa$ should quantify confidence in the prediction of $\hat{y}$ for the input $x$, based on signals from model $f$. This function should induce a partial order over instances in $\mathcal{X}$.

The most common and well-known $\kappa$ function for a classification model $f$ (with softmax at its last layer) is its softmax response values – $\kappa(x, \hat{y}|f) \triangleq f(x)_{\hat{y}}$ (Cordella et al., 1995; De Stefano et al., 2000) – which is also widely accepted as a baseline in the OOD literature (Hendrycks & Gimpel, 2017; Hendrycks et al., 2021; Berger et al., 2021; Shalev et al., 2018). While this is the primary $\kappa$ we evaluate for the sake of simplicity, various other $\kappa$ functions, which are also utilized for OOD detection, exist. To name a few: *Out-of-distribution detector for neural networks* (ODIN) (Liang et al., 2018), *Monte-Carlo dropout* (MC dropout) (Gal & Ghahramani, 2016), *Mahalanobis distance* (Lee et al., 2018), and more. Although many of these methods use the direct output from $f$, $\kappa$ could be a different model unrelated to $f$ and unable to affect its predictions.

$\kappa$ functions can be evaluated by the quality of the partial order they induce over instances in $\mathcal{X}$. For every two random samples $(x_1, y_1), (x_2, y_2) \sim P(\mathcal{X}, \mathcal{Y})$, and given that $x_1$ belongs to an OOD label and that $x_2$ belongs to an ID label, the *detection* (or *ranking*) performance of $\kappa$ is defined as the probability that $\kappa$ ranks $x_2$ higher than $x_1$:

$$\mathbf{Pr}[\kappa(x_1, \hat{y}_1|f) < \kappa(x_2, \hat{y}_2|f) \mid x_1 \sim P(\mathcal{X}, \mathcal{Y}_{\text{OOD}}) \wedge x_2 \sim P(\mathcal{X}, \mathcal{Y}_{\text{ID}})] \tag{1}$$

The *Area Under the Receiver Operating Characteristic* (AUROC or AUC) metric is often used to measure the performance of OOD detection. When ID samples are counted as true positives and OOD samples are counted as false positives, AUROC, in fact, equals the probability in Equation (1) (Fawcett,

2006) and thus is a proper metric to measure OOD detection in classification. See Appendix A for evaluating $\kappa$ functions in an ID setting.

## 3 CONSTRUCTING A MODEL-SPECIFIC CLASS-OUT-OF-DISTRIBUTION BENCHMARK

We first choose a dataset that contains samples from a large set of OOD labels (e.g., labels from ImageNet-21k that are not included in ImageNet-1k). Ideally, this OOD dataset should consist of OOD labels representing labels the model may encounter when deployed. Any large dataset could be used for the purpose of benchmarking performance on C-OOD by splitting it according to labels into an ID component, i.e., the labels on which the model trains, and into an OOD component, i.e., the labels on which the model is exclusively tested.

We now introduce a novel framework for generating C-OOD benchmarks with a controllable degree of *severity*, which could be thought of as the difficulty level of the data. Algorithm 1 summarizes our proposed technique. Let $\mathcal{Y}_{\text{OOD}}$ be a large set of OOD classes (e.g., labels from ImageNet-21k that are

---

**Algorithm 1** Generating C-OOD benchmarks

1: **function** GENERATE_BENCHMARK($f, \kappa, \mathcal{Y}_{\text{OOD}}, group\_size = |\mathcal{Y}_{\text{ID}}|$)
2:     **for** $\bar{y} \in \mathcal{Y}_{\text{OOD}}$ **do**
3:         Split all samples of class $\bar{y}$ into two sets: $c_{est}^{\bar{y}}$ and $c_{test}^{\bar{y}}$
4:         Set the severity score of class $\bar{y}$ to be: $s(\bar{y}|f, \kappa) = \frac{1}{|c_{est}^{\bar{y}}|} \sum_{x \in c_{est}^{\bar{y}}} \kappa(x|f)$.
5:         Insert the class and its score $(\bar{y}, s(\bar{y}|f, \kappa))$ into $classes\_array$
6:     Sort $classes\_array$ in ascending order by each OOD class' score $s(\bar{y}|f, \kappa)$
7:     **for** $i < |\mathcal{Y}_{\text{OOD}}| - group\_size$ **do**         ▷ Sliding window of size $group\_size$
8:         $grp\_array[i] = classes\_array[i : i + group\_size]$
9:     **for** $i < 11$ **do**         ▷ Select groups in different percentiles to serve as benchmarks
10:         $sev\_benchmark[i] = \{x \mid x \in c_{test}^{\bar{y}} \text{ s.t. } \bar{y} \in grp\_array[j] \text{ and } j = \lceil \frac{i}{10} \cdot |grp\_array| \rceil\}$
11:     **return** $sev\_benchmark$

---

not included in ImageNet-1k), and let $s_{f,\kappa}(\bar{y})$ be a *severity score*, defined as the average confidence given by $\kappa$ to samples from class $\bar{y} \in \mathcal{Y}_{\text{OOD}}$. This score reflects the level of difficulty faced by the model $f$ and its $\kappa$ function when detecting instances from class $\bar{y}$. When considering ID instances we expect $\kappa$ to give high values for highly confident predictions. Therefore, the larger $s(\bar{y}|f, \kappa)$ is, the harder it is for $\kappa$ to detect the OOD class $\bar{y}$ among ID classes. We estimate $s(\bar{y}|f, \kappa)$ for each class in the OOD dataset (e.g., ImageNet-21K) using a set of samples from the class (denoted by $c_{est}^{\bar{y}}$), while keeping a disjoint set of samples from the same class to be used for testing (denoted by $c_{test}^{\bar{y}}$). Using $s$ we sub-sample groups of classes (severity levels) from $\mathcal{Y}_{\text{OOD}}$, with increasing severity such that severity level $i \in [0, 10]$ is the $i^{th}$ percentile of all severity levels.

To achieve this, we first estimate the severity score for each class $\bar{y}$ in our OOD dataset for our model and its confidence function $(f, \kappa)$, as follows:

$$s(\bar{y}|f, \kappa) = \frac{1}{|c_{est}^{\bar{y}}|} \sum_{x \in c_{est}^{\bar{y}}} \kappa(x|f).$$

We group the OOD classes into different groups, and choose the size of each group $G$ to be the same as $|\mathcal{Y}_{\text{ID}}|$, the number of labels in the ID dataset (e.g., in ImageNet we choose it to be 1000 classes). The number of possible groups of labels from $\mathcal{Y}_{\text{OOD}}$ could be huge (in ImageNet, for example, the number of possible groups of size 1000 from the 20,000 OOD classes is about $\binom{20,000}{1000} = 2.5 \times 10^{1722}$), so instead of going over every possible group of classes, we sort the classes by their severity scores and then use a sliding window of size $|\mathcal{Y}_{\text{ID}}|$ to define $|\mathcal{Y}_{\text{OOD}}| - |\mathcal{Y}_{\text{ID}}| + 1$ groups of classes with increasing severity (see Figure 2). This method for reducing the number of considered groups of classes was chosen because it groups OOD classes with similar severity scores together.

Next, we choose the groups that correspond to the percentiles $\{10 \cdot i\}_{i=0}^{i=10}$ in the array of sorted groups. Finally, we construct the C-OOD benchmark for each severity level $i$ from the set of test

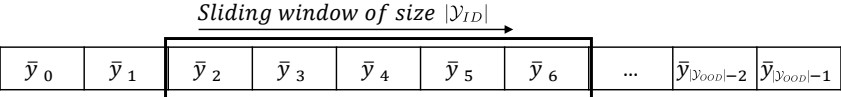

Figure 2: We define $|\mathcal{Y}_{\text{OOD}}| - |\mathcal{Y}_{\text{ID}}| + 1$ groups of classes with increasing severity by sorting all OOD classes $\bar{y}_i \in \mathcal{Y}_{\text{OOD}}$ by their severity scores $s(\bar{y}|f, \kappa)$, and then use a sliding window of size $|\mathcal{Y}_{\text{ID}}|$ to choose the considered groups.

samples $c_{test}^{\bar{y}}$ of all classes in group $i$. This procedure for choosing groups allows us to interpret the severity levels using percentiles. For example, severity level 5 contains classes that match the median severity among the considered groups. Thus, the performance evaluated on the benchmark for severity 5 corresponds to the performance of the model on samples with a median detection difficulty.

The resulting benchmark is tailored to the evaluated model, since the latter was used to generate it and, therefore, can be used to measure its specific performance. In Appendix B we further argue why our framework can be used to compare C-OOD detection performance of different models.

## 4 CONSTRUCTING BENCHMARKS FOR IMAGENET CLASSIFIERS

To use ImageNet-21k as an OOD dataset, we first filter out undesired labels. Since ImageNet-21K contains the ID dataset (ImageNet-1K), the first step is to remove the ID classes from the OOD dataset. Next, we remove all classes that are hypernyms or hyponyms of classes in ImageNet-1K because it might be inaccurate to include them as an OOD class. For example, ImageNet-1K contains the class "brown bear" and ImageNet-21K has the class "bear", which is a hypernym for "brown bear" so it would not be accurate to include "bear" in a C-OOD detection test. We furthermore filter OOD classes that, together with an ID class, either comprise the same object or are a component of the other one. This is due to most images in the dataset containing both components as parts of the whole object (e.g., "pool ball" from ImageNet-1k and "pool table" from ImageNet-21k). We also filter out classes that are practically identical, even though they possess WordNet id numbers that are different (e.g., "hen" is found twice as two distinct classes, with id n01514859 in ImageNet-1k and id n01792640 in ImageNet-21k). Since each class in the ImageNet-1k validation set has 50 samples, we set the number of testing samples for each C-OOD class to be 50 as well $|c_{test}^{\bar{y}}| = 50$. In addition, We set the estimation set for each class to be 150 $|c_{est}^{\bar{y}}| = 150$. Overall, this means that each OOD class must have at least 200 samples. Accordingly, we remove classes with less than 200 samples. For classes with more than 200 samples we randomly select 200 samples and remove the rest.

While the above filtering choices are trivial and suitable for most tasks, two additional filtering options are dependent on the task and its definition of two objects being considered identical. The first option concerns animal classes that might appear to be very similar but have a biological difference such that an expert could distinguish between the two. A good example of this can be observed in Figure 3, depicting the ImageNet-1k class of Monarch butterflies and the ImageNet-21k class of Viceroy butterflies, which are both distinct species of butterflies. The similarity is so remarkable that scientists believe they have evolved to mimic one another to repel common predators (Ritland & Brower, 1991). This mimicry does not only fool predators and the untrained eye: *all* models studied in this paper classified more than 50% of Viceroy samples as a Monarch butterfly. The fact that such

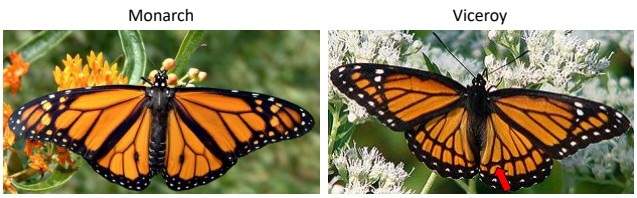

Figure 3: While both butterflies appear very similar, a Viceroy can be distinguished from a Monarch by a black line crossing its postmedian hindwing. The red arrow on the Viceroy image indicates this black line.

classes are biologically different led us to keep them in the test set by default and serve as extremely

hard OOD classes. Our code, however, allows users to disable such classes easily, since some tasks might permit such similar classes to be classified as the same.

The second option concerns inanimate objects created by humans that might appear very similar but are, by definition, distinct from one another and are used differently. An example of two such classes

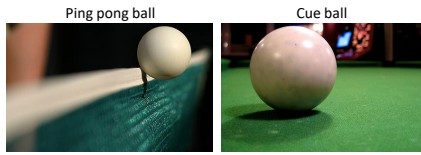

Figure 4: While both balls appear similar, they are distinguished by their different uses.

is shown in Figure 4, depicting a cue ball used for billiard games and a ping pong ball. Both are strikingly similar, and we believe a person completely unfamiliar with one of the games might easily confuse the two, if all they had were the images. Our code can be configured easily to either exclude or include such classes.

After completing the filtering as described above, the remaining classes were used in the process described in Section 3 as the set of OOD classes $\mathcal{Y}_{\text{OOD}}$, with ImageNet's validation set being the set of ID classes $\mathcal{Y}_{\text{ID}}$. Our code allows the generation of C-OOD benchmarks for any ImageNet classification model and its $\kappa$ confidence scoring function. Moreover, we ran the process ourselves for 525 models pretrained on ImageNet, taken from the torchvision (0.10) and "timm" (0.4.12) repositories (Paszke et al., 2019; Wightman, 2019), with softmax as $\kappa$. For these models, the benchmarks are ready to be used by the community without further preparations being necessary.

## 5 PERFORMANCE ANALYSIS

Having generated C-OOD benchmarks using the above technique for 525 different models , in this section we analyze the results. We first focus on results obtained when setting the confidence function $\kappa$ to be the softmax response, as it is widely accepted as a baseline in the OOD literature (Hendrycks & Gimpel, 2017; Berger et al., 2021). We then evaluate additional $\kappa$ functions such as ODIN, entropy and MC dropout. Our analysis leads to several interesting insights.

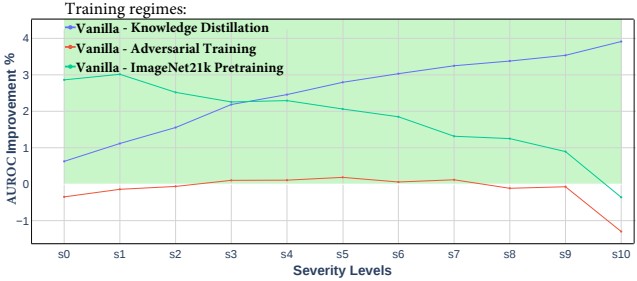

Figure 5: The mean relative improvement when using different training regimes (distillation, pretraining etc.). The shaded green area indicates the area of positive improvement.

1) **Knowledge distillation improves C-OOD detection**. We measured C-OOD detection improvement (measured in AUROC) when using different training regimes to explore whether a certain method consistently contributes to detection performance. Results are depicted in Figure 5. To make a fair comparison, we only compare pairs of models such that both models have identical architecture and training regimes, with the exception of the method itself being evaluated (e.g., training with or without knowledge distillation). Of all training regimes (knowledge distillation, adversarial training (Goodfellow et al., 2015), pretraining on ImageNet-21k, see below), knowledge distillation had the most significant impact in most severity levels $s > 3$. In Galil et al. (2023) we also find that among these training regimes, knowledge distillation is the best booster of uncertainty estimation performance in an in-distribution setting. Next, we find that ImageNet21k pretraining also improves performance, and is more beneficial to performance than knowledge distillation in low levels of

severity $s \leq 3$. Note that this observation could not have been achieved with simplified benchmarks (e.g., ImageNet-O). Our new framework allows for such observations thanks to the division of the benchmarks into different levels of severity. Finally, it is not surprising that adversarial training is irrelevant to C-OOD detection.

2) **A subset of ViTs achieves the best C-OOD detection performance**, both in absolute terms and per-model size (# parameters, see Figure 9 in Appendix C). Several training regimes (including the original regime from the paper introducing ViT) result in ViTs that outperform all other architectures and training regimes in terms of C-OOD detection, e.g., Dosovitskiy et al. (2021); Steiner et al. (2022); Chen et al. (2022); Ridnik et al. (2021). Further research into other training regimes, however, reveals that not all training regimes result in superb performance (Touvron et al., 2021; 2022; Singh et al., 2022; Paszke et al., 2019), even when a similar amount of data is introduced into the training. We also find that the same successful subset of ViTs outperforms any other model in terms of uncertainty estimation performance in an in-distribution setting in Galil et al. (2023). These observations warrant additional research with the hope of either training more robust ViTs or transferring the unidentified ingredient of the successful subset of ViTs into other models.

3) **The language–vision CLIP model achieves good zero-shot C-OOD detection performance for low severity levels**. CLIP (Radford et al., 2021) enables zero-shot classification and produces an impressive performance. We find it is also good at C-OOD detection (especially in severity levels lower than 6), without needing any training or fine-tuning with regard to the dataset. This observation is significant because it means CLIP could be used as a zero-shot C-OOD detection algorithm without the need to train on the ID classes. This also allows the user to change the definition of which classes are considered ID in a flexible manner without the need to retrain the detector. To the best of our knowledge, we are the first to make the observation that CLIP can serve as a capable zero-shot detector on its own, without further training, additional components, or knowledge of the possible OOD classes in advance. For more details, see Appendix D.

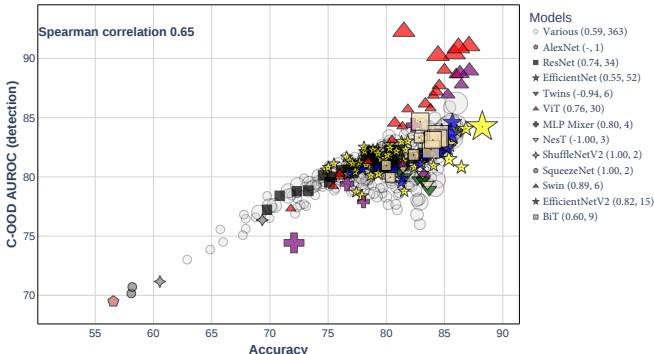

Figure 6: Architecture accuracy vs. mean C-OOD AUROC performance. In the legend, the pair of numbers next to each architecture name corresponds to the Spearman correlation and the number of networks tested from that architecture family (most samples are too small to draw any specific conclusions). Accuracy appears to have a high correlation with the C-OOD detection performance, with a Spearman correlation of 0.65.

4) **Accuracy is the factor most correlated with C-OOD detection**. We observe that accuracy is typically a good indicator of the model's performance in C-OOD detection at most severity levels $[s_0 - s_8]$, with Spearman correlation values in the range of $[0.6, 0.73]$ at those levels (see Figure 12 in Appendix E). The scatter plot in Figure 6 shows the relationship between the architecture accuracy and its C-OOD detection performance. When grouping the networks by architecture, we notice that most architectures also follow this trend. When measuring the correlation between AUROC and accuracy among only the 20% most accurate models, however, the Spearman correlation drops to a range of $[0.34, 0.43]$ (see Figure 13 in Appendix E).

5) **In-distribution ranking performance is positively correlated with C-OOD detection**. The next best indicative factor correlated with C-OOD detection performance after accuracy is the model's in-distribution ranking performance ("ID AUROC", see Appendix A), with Spearman correlation values in the range of $[0.4, 0.5]$. When measuring the correlation between AUROC and ID AUROC

among only the 20% most accurate models, however, the Spearman correlation increases to a range of $[0.54, 0.77]$; see Appendix E for more details.

6) **Most OOD classes appear in every severity level $i \in [0, 10]$ for at least one model**, with the exception of some classes that appear to reach severity level 10 for most or even all models (e.g., Viceroy Butterfly, depicted in Figure 3 in Section 4). This observation suggests that "OOD hardness" is usually subjective, and changes greatly across different models.

7) **The ranking of the best C-OOD detection models tends to remain similar across severity levels**. This means that when selecting the best model for deployment, it is usually enough to observe its performance on only a few severity levels; see Appendix F. Note that this conclusion is only true when leaving the $\kappa$ confidnece function fixed (see below).

8) **ODIN offers significant improvements over softmax for most models**. In addition to evaluating with softmax as the $\kappa$ confidence function, we evaluate a few additional methods to serve as $\kappa$ functions: ODIN, entropy, MC dropout and "max-logit" (not applying softmax). For each model $f$ and $\kappa$ we re-ran the algorithm described in Section 3 to benchmark $(f, \kappa)$ (we do this because using the same C-OOD groups produced when using softmax might give an unfair advantage to other $\kappa$ functions); see Appendix G for more technical details.

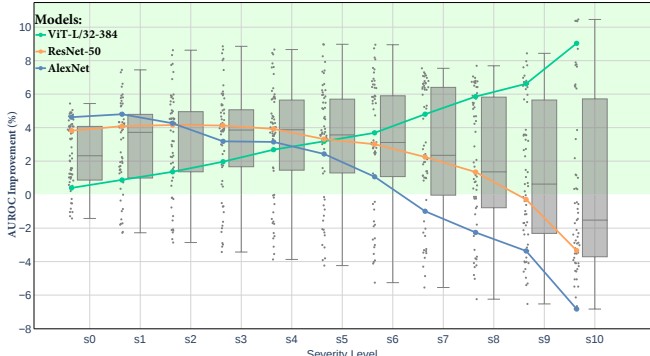

Figure 7: Relative improvement gain in C-OOD detection performance when using ODIN instead of softmax. Each point represents an evaluated model. The green shaded area indicates the area of positive improvement.

Figure 7 shows each model's improvement when using ODIN rather than softmax, from which it is visible that the improvement has a high variance: some models benefit significantly from using ODIN, while it is detrimental to other models. Furthermore, whether or not a model benefits from ODIN changes across different levels of severity. For example, applying ODIN instead of softmax to ViT-L/32-384 barely improves detection when at severity level 0 (AUROC improves by $0.4\%$), but it significantly improves its detection as the severity level increases (for severity level 10, AUROC improves by $9\%$). Other models' detection performance, on the other hand, may decrease as severity increases (see Figure 7 for examples). These facts suggest that the pair of (model, $\kappa$) needs to be considered with respect to the task and severity level relevant to it. Moreover, it may be that the $\kappa$ function hyperparameters need to be optimized specifically for the desired severity level.

9) **Not applying softmax can improve some models significantly, although most are harmed by it**. Figure 16 in Appendix G depicts the effect of not applying softmax, which we dub "max-logit". While most models are harmed by using max-logit instead of softmax, some models are significantly benefited. ViTs, which already outperform all other models, perform significantly better when softmax is not applied, with ViT-L/32-384 improving by $10.6\%$. It is worth mentioning that of all the (model,$\kappa$) pairs evaluated in this paper, ViT-L/32-384 applied with max-logit achieve the best detection performance. Interestingly, regardless of the $\kappa$ function evaluated, ViT-L/32-384 demonstrated the best detection performance. In Figure 8, we plot its performance across all severity levels using each of the $\kappa$ functions we consider. Also, as noted in Appendix G, the hyperparameters used for ODIN when applied to ViT were not optimized specifically to it. Performance by using ODIN may improve beyond max-logit with model-specific optimization. Observing that max-logit could be so beneficial for a subset of models while being harmful to most other models was made possible thanks to the scale of our study.

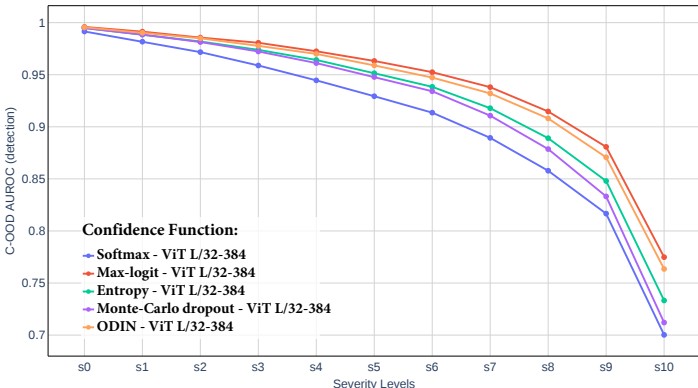

Figure 8: OOD detection performance of ViT-L/32-384, the best model evaluated using each of the $\kappa$ functions we consider.

10) **Using entropy as a confidence function $\kappa$ improves C-OOD detection performance** in most cases. We compare the performance gain from switching to using entropy instead of the softmax score. The results are depicted in Figure 17 in Appendix G. We note that, in most cases, using entropy improves the detection performance. [1]

11) **MC dropout improves detection, especially for low levels of severity**. We evaluate MC dropout Gal & Ghahramani (2016) in the context of C-OOD detection. We use 30 dropout-enabled forward passes. The mean softmax score of these passes is calculated and then a predictive entropy score is used as the final uncertainty estimate. The improvements when using MC dropout instead of softmax across all severity levels are depicted in Figure 18 in Appendix G using box plots. We find that MC dropout improves performance, especially so at lower levels of severity. The improvement becomes less significant as severity increases. Similar to ODIN, MC dropout seems to improve some models more significantly at lower severity levels (e.g., MobileNets (Howard et al., 2019)) , while other models are improved more significantly by MC dropout at higher severity levels (e.g., ViTs). We further analyze MC dropout and recall that it comprises two main components: (a) dropout-enabled forward passes and (b) entropy of the mean probability vector from the forward passes. To test which component contributes the most to the perceived gains, we compare the C-OOD detection performance when using MC dropout to the C-OOD detection performance when using just entropy (with no multiple dropout-enabled forward passes). The results of this comparison are plotted in Figure 19 in Appendix G. We find that MC dropout slightly improves upon entropy at most severity levels, especially at lower ones, with few outliers being either significantly improved or harmed.

## 6 Concluding remarks

We introduced a novel approach to benchmarking the performance of classifiers in detecting C-OODs. In contrast to existing techniques, the proposed method allows for unbiased measurements against specific models or confidence functions. A key feature of the proposed benchmarking procedure is that it allows for graded measurements of class out-of-distribution levels of severity. Using this property, we can identify trends in detection robustness that are otherwise impossible to detect. In addition to opening new avenues for future research, the proposed method can be used to draw more precise conclusions about the performance of various models and detection techniques.

Using our new benchmarking procedure, we offered numerous interesting observations that merit further investigation into how to improve C-OOD detection. Among the interesting questions raised is why is knowledge distillation beneficial to boosting detection performance, and how can we enhance its robustness to C-OODs? What can we learn from the architectures that were inclined to perform well in C-OOD detection, such as ViT and CLIP? Finally, could detection methods be crafted and optimized for specific severity levels, or can they be modified to be so by changing a hyperparameter?

---

[1]Entropy is maximal when the distribution given by the model for $P(y|x)$ is uniform, which implies high uncertainty. To convert entropy into a *confidence signal*, which should increase as the uncertainty decreases, we use negative entropy.

## ACKNOWLEDGMENTS

This research was partially supported by the Israel Science Foundation, grant No. 710/18.

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

## A   Defining in-distribution AUROC

We follow Galil et al. (2023) in defining in-dsitribution AUROC ("ID AUROC"). ID AUROC is defined similarly to Equation 1, but discriminating between correct and incorrect predictions instead of discriminating between ID and OOD instances.

For every two random samples $(x_1, y_1), (x_2, y_2) \sim P(\mathcal{X}, \mathcal{Y})$ and given that $\ell(f(x_1), y_1) > \ell(f(x_2), y_2)$, the *ranking* performance of $\kappa$ is defined as the probability that $\kappa$ ranks $x_2$ higher than $x_1$:

$$\mathbf{Pr}[\kappa(x_1, \hat{y}|f) < \kappa(x_2, \hat{y}|f)|\ell(f(x_1), y_1) > \ell(f(x_2), y_2)] \tag{2}$$

When the 0/1 loss is in play, it is known that AUROC in fact equals the probability in Equation (2) (Fawcett, 2006) and thus is a proper metric to measure ranking in classification (AKA ID AUROC or discrimination).

## B    COMPARING MODELS' PERFORMANCE USING OUR FRAMEWORK

The proposed framework allows for a fair comparison of models in terms of model-specific **difficulty**, rather than a fixed set of OOD classes chosen according to some (possibly arbitrary) criterion. This is because the framework evaluates each model's performance on tailored benchmarks. This approach provides a more accurate representation of the model's own performance. As the famous quote goes, "You can't judge a fish by its ability to climb a tree". Rephrasing this quote to adapt it to our discussion: if we want to compare a fish with a monkey on what is hardest for each of them, we should judge the fish by its ability to climb a tree and the monkey's ability to swim (although we are aware that some monkeys can swim). Our framework constructs specialized tests for both.

That being said, by considering the construction of severity levels (per model), it is possible (neglecting estimation error of the estimation sets $c_{est}^{\bar{y}}$) to compare the performance of two models specifically for the classes populating their maximal severity (severity 10):

(1) Suppose that model $\mathcal{A}$ has better performance (AUROC) on its own group $Z$ of hardest classes (severity 10) than model $\mathcal{B}$'s performance on its own severity 10 classes, denoted $K$. Assume that $K$ does not equal $Z$ (otherwise we are done). Thus, $\text{AUROC}(\mathcal{A}, Z) > \text{AUROC}(\mathcal{B}, K)$.

(2) By construction of severity groups, for every set of classes $R \neq Z$, $\text{AUROC}(\mathcal{A}, R) \geq \text{AUROC}(\mathcal{A}, Z)$ (since $Z$ is the set of hardest classes for model $\mathcal{A}$). This holds true for any set of classes $R$, including the set $K$. Therefore, $\text{AUROC}(\mathcal{A}, K) \geq \text{AUROC}(\mathcal{A}, Z)$.

By combining (1) and (2) we get that $\text{AUROC}(\mathcal{A}, K) \geq \text{AUROC}(\mathcal{A}, Z) > \text{AUROC}(\mathcal{B}, K) \Rightarrow \text{AUROC}(\mathcal{A}, K) > \text{AUROC}(\mathcal{B}, K)$, meaning that for the same set of classes $K$, model $\mathcal{A}$ performs better than model $\mathcal{B}$.

A "mirror" argument could be crafted to compare the models' performance on the classes populating their minimal severity (severity 0).

## C    PER-SIZE PERFORMANCE COMPARISON

The scatter plot in Figure 9 shows the relationship between the # of architecture parameters and its C-OOD AUROC performance. Overall, there is a moderate Spearman correlation of 0.45 between #parameters and the C-OOD performance when considering all tested networks. When grouping the networks by architecture families, however, we see that some architectures have high correlation between their model size and their C-OOD AUROC. Architecture families that exhibit this behavior are, for example, ViTs, Swins, EffecientNetV2 and ResNets whose correlations are 0.91, 0.94, 0.89, and 0.79, respectively. Other families exhibit moderate correlations, e.g., EffecientNet(V1) with a 0.47 Spearman correlation. Some architectures, on the other hand, have strong negative correlation, e.g., Twins Chu et al. (2021), NesT Zhang et al. (2020) and Res2Net Gao et al. (2021), whose correlations are -0.94,-1.0, and -0.85, respectively.

Additionally, we note that the subset of ViT models mentioned in Section 5 are also the best even when considering a model size limitation.

## D    ZERO-SHOT C-OOD DETECTION WITH CLIP

To evaluate CLIP on ImageNet, we first prepare it following the code provided by its authors (https://github.com/openai/CLIP): The labels of ImageNet-1k are encoded into normalized embedding vectors. At inference time, the incoming image is encoded into another normalized embedding vector. A cosine similarity is then calculated between each label-embedding vector and the image-embedding vector. The highest similarity score is then taken as the confidence score for that prediction.

To evaluate CLIP's C-OOD performance, we re-run the algorithm described in Section 3 to benchmark $(CLIP, \kappa_{cosine\ similarity})$. The best-performing instance of CLIP (ResNet-50x64) outperforms 96%

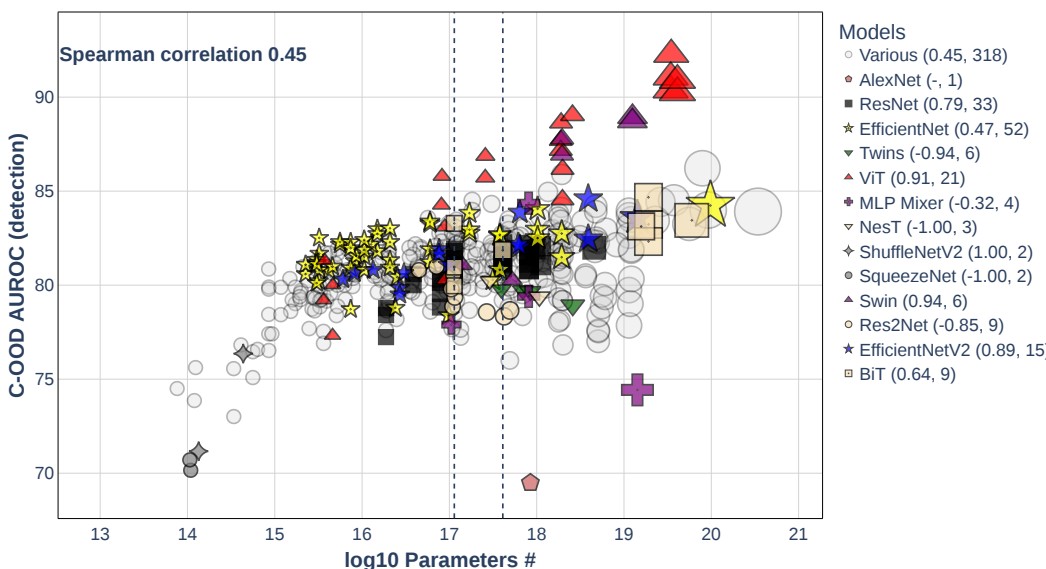

Figure 9: Number of architecture parameters vs. C-OOD AUROC performance at severity level 5 (median severity). The pair of numbers next to each architecture name in the legend corresponds to its Spearman correlation and the number of models tested from that architecture (family), respectively. Note that specific ViT transformers are also the best when considering a model size limitation. Vertical lines indicate the sizes of ResNet-50 (left vertical line) and ResNet-101 (right vertical line).

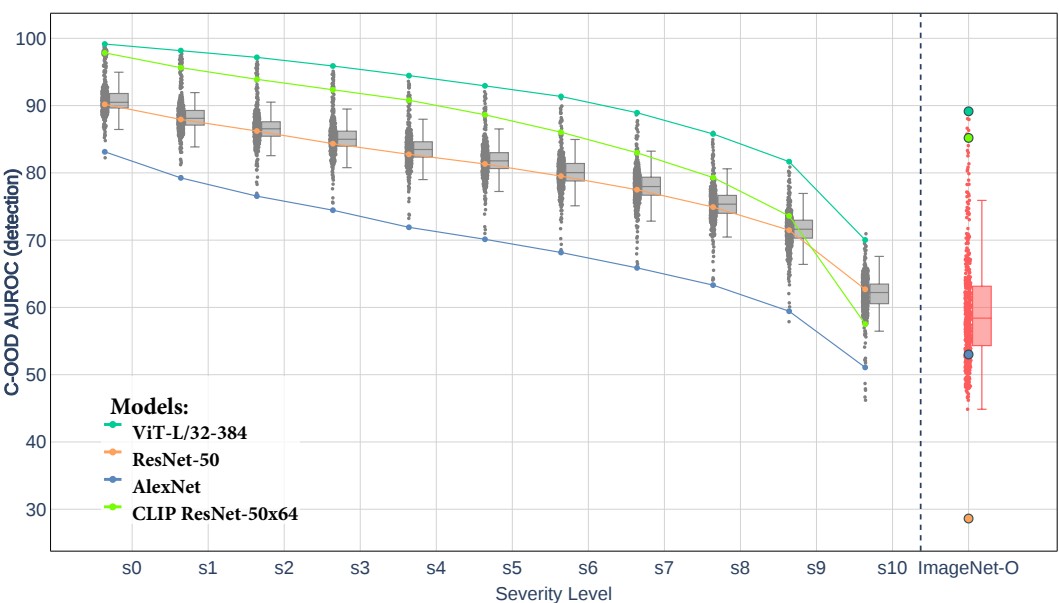

Figure 10: The same graph as in Figure 1, but with an additional lime-colored curve for CLIP ResNet-50x64. Note that as severity levels increase, CLIP's detection advantage is greatly reduced.

of all other models (measured by its mean AUROC over all severity levels). In Figure 10 we visualize this CLIP's performance across all severity levels, in comparison to all other models. Interestingly, CLIP's relative advantage over other models decreases as the severity increases, and at severity 10, it is even lower than the median. The same is observed in Figure 11 which depicts a comparison between three identical ResNet-50 models that were trained with three different training regimes, one of them being CLIP. CLIP outperforms its competition up to severity 6 (with a significant margin in lower

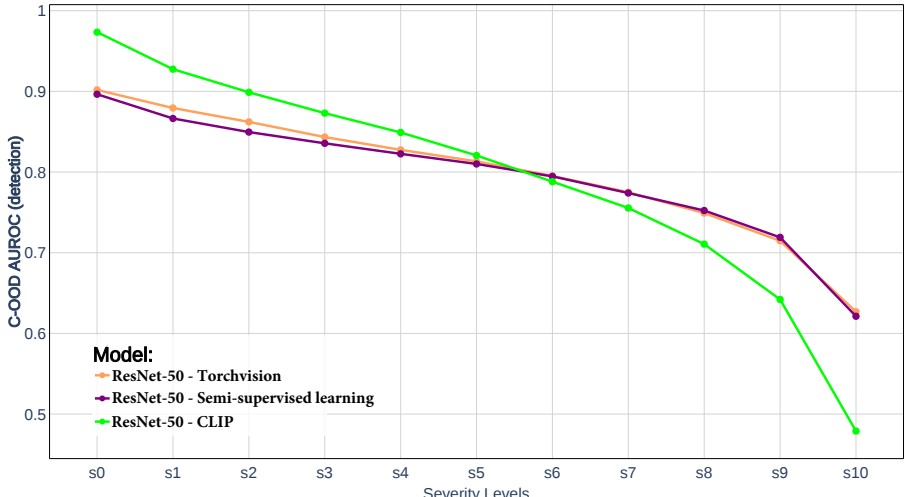

Figure 11: A comparison of three identical ResNet-50 models trained with different training regimes: (1) The orange-colored curve represents a ResNet-50 model trained on ImageNet-1k with Torchvision's recipe; (2) the purple-colored curve represents a ResNet-50 model trained with a semi-supervised regime (Yalniz et al., 2019); and (3) the lime-colored curve represents a ResNet-50 trained with CLIP.

severity levels), and then underperforms. We hypothesize the degradation in CLIP's performance for higher severity levels happens due to an increase in the number of OOD classes that are descriptively similar to ID classes at higher levels of severity. For example, when examining different types of butterflies from Figure 3, the string text of "monarch butterfly" is very similar to the string text of "viceroy butterfly", simply due to both sharing the word "butterfly". Other butterflies that are less visually similar might be "confused" by CLIP and classified as monarch butterflies, simply because they are also defined as butterflies, making their cosine similarity with the text "monarch butterfly" higher. Common image classifiers, on the other hand, may confuse different butterflies if they appear visually similar and share many distinguishable features, but are not affected by the fact both classes are defined as "butterflies".

We also observe that while CLIPs with a confidence function $\kappa_{cosine\ similarity}$ perform very well at C-OOD detection, their ID ranking is worse than other models. Using softmax and\or adding a linear-probe (as described in Radford et al. (2021)) improves ID ranking significantly, but results in mediocre C-OOD detection performance. We believe that this suggests the multimodal nature of CLIP is a crucial component of its C-OOD detection performance, and that the scaling effect of softmax hinders the partial order induced on OOD and ID instances.

In Fort et al. (2021), it was suggested that CLIP be used as a zero-shot OOD detection algorithm. Their suggested method, however, requires knowledge of the possible OOD classes in advance. The authors of Esmaeilpour et al. (2022) suggested to use an additional captioning model, which is fine-tuned on some large dataset (which hopefully contains knowledge of the OOD classes that might emerge during inference), instead. Our suggested approach, in contrast, requires no knowledge, no fine-tuning and no models other than CLIP itself.

# E   CORRELATIONS OF VARIOUS FACTORS WITH C-OOD DETECTION PERFORMANCE

We searched for factors that could be indicative of or correlated with good performance in C-OOD detection. To this end, we measure the correlations of various factors with the C-OOD detection AUROC performance across all levels of severity. The results can be seen in the graphs in Figure 12. We observe that accuracy is typically a good indicator of the model's performance in C-OOD detection at most severity levels ($s_0 - s_8$), with Spearman correlation values in $[0.6, 0.73]$ at those levels (see Figure 12). When measuring the correlation between AUROC and accuracy among only

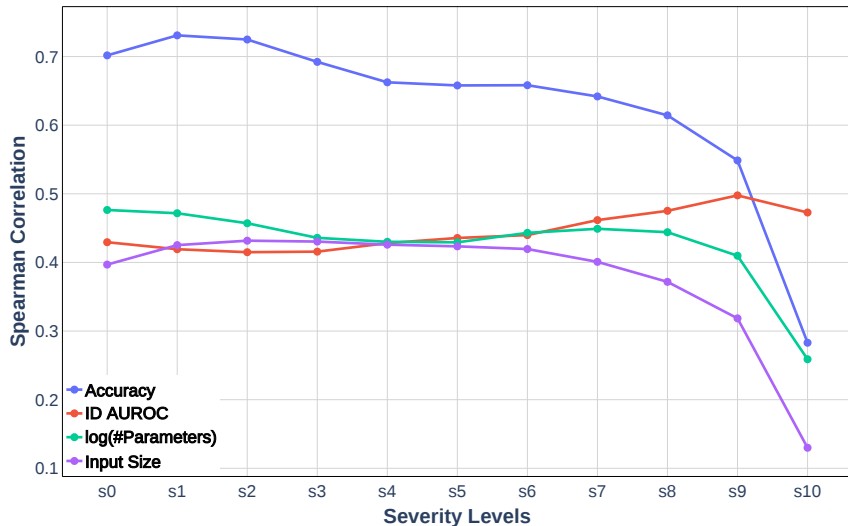

Figure 12: Spearman correlations between C-OOD detection AUROC and Accuracy, ID-AUROC, #parameters, input size, and embedding size across all severity levels.

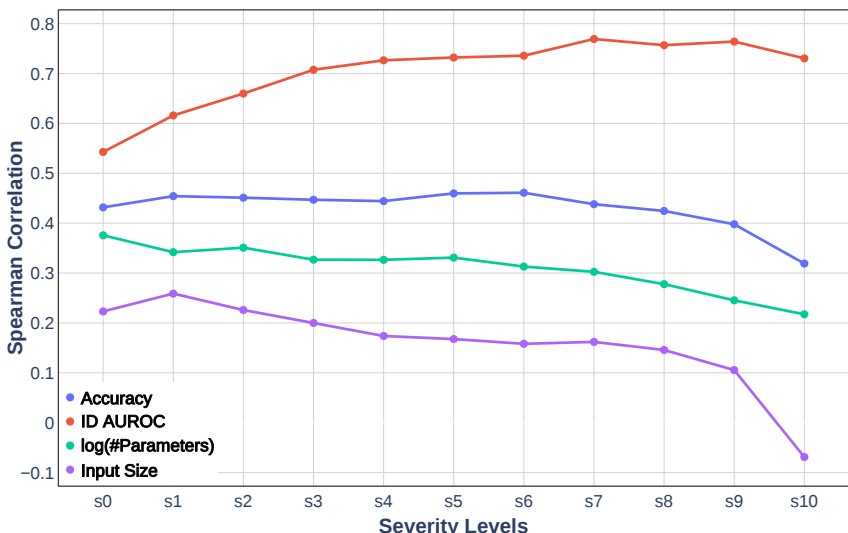

Figure 13: Spearman correlations between C-OOD detection AUROC and Accuracy, ID-AUROC, #parameters, input size, and embedding size across all severity levels, **among only the 20% most accurate models**.

the 20% most accurate models, however, the Spearman correlation drops to a range of $[0.34, 0.43]$ (see Figure 13).

The next best indicative factors are the ID ranking performance ("ID AUROC"), number of parameters, and the input image size (moderate correlations). Finally, the embedding size is only weakly correlated.

Figure 14 shows a scatter plot of in-distribution ranking performance and C-OOD detection performance of all evaluated models. The overall Spearman correlation is 0.43. The legend indicates correlations obtained by specific architecture families. Interestingly, ID AUROC exhibits slightly increasing correlation up to severity $s_9$, and at $s_{10}$ becomes the most indicative factor for C-OOD detection performance. In contrast, all other investigated factors lose their indicative power at the highest severity levels ($s_9, s_{10}$). Moreover, when measuring the correlation between AUROC and ID AUROC among only the 20% most accurate models, the Spearman correlation increases to a range

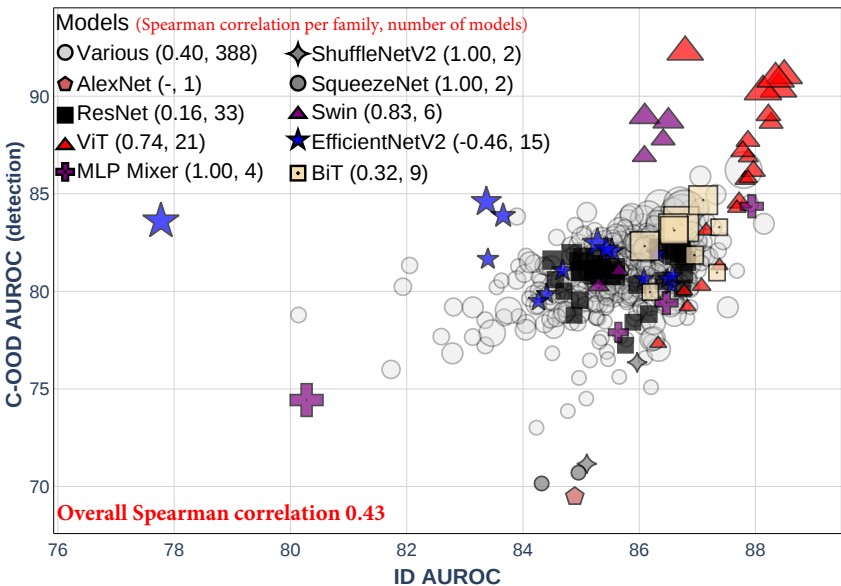

Figure 14: The x-axis represents ID ranking performance (measured by AUROC), and the y-axis represents C-OOD detection performance in severity 5 (higher is better). The legend indicates correlations, by specific architecture families, with the number on the right representing sample size, and the one on the left representing the correlation between ID ranking and detection.

of $[0.54, 0.77]$, making it the most indicative factor for C-OOD detection among such models (see Figure 13).

# F   CORRELATION BETWEEN RANKINGS OF MULTIPLE SEVERITY LEVELS

|      | s0    | s1    | s2    | s3    | s4    | s5    | s6    | s7    | s8    | s9    | s10   |
|------|-------|-------|-------|-------|-------|-------|-------|-------|-------|-------|-------|
| s10  | 0.376 | 0.416 | 0.461 | 0.513 | 0.549 | 0.576 | 0.603 | 0.634 | 0.675 | 0.812 | 1.0   |
| s9   | 0.69  | 0.739 | 0.779 | 0.824 | 0.86  | 0.888 | 0.913 | 0.939 | 0.961 | 1.0   | 0.812 |
| s8   | 0.787 | 0.846 | 0.883 | 0.916 | 0.944 | 0.963 | 0.979 | 0.989 | 1.0   | 0.961 | 0.675 |
| s7   | 0.815 | 0.879 | 0.915 | 0.946 | 0.969 | 0.983 | 0.992 | 1.0   | 0.989 | 0.939 | 0.634 |
| s6   | 0.831 | 0.9   | 0.938 | 0.966 | 0.983 | 0.993 | 1.0   | 0.992 | 0.979 | 0.913 | 0.603 |
| s5   | 0.846 | 0.918 | 0.955 | 0.98  | 0.993 | 1.0   | 0.993 | 0.983 | 0.963 | 0.888 | 0.576 |
| s4   | 0.854 | 0.931 | 0.968 | 0.991 | 1.0   | 0.993 | 0.983 | 0.969 | 0.944 | 0.86  | 0.549 |
| s3   | 0.882 | 0.955 | 0.985 | 1.0   | 0.991 | 0.98  | 0.966 | 0.946 | 0.916 | 0.824 | 0.513 |
| s2   | 0.928 | 0.986 | 1.0   | 0.985 | 0.968 | 0.955 | 0.938 | 0.915 | 0.883 | 0.779 | 0.461 |
| s1   | 0.965 | 1.0   | 0.986 | 0.955 | 0.931 | 0.918 | 0.9   | 0.879 | 0.846 | 0.739 | 0.416 |
| s0   | 1.0   | 0.965 | 0.928 | 0.882 | 0.854 | 0.846 | 0.831 | 0.815 | 0.787 | 0.69  | 0.376 |

Figure 15: Spearman correlation between the rankings of the models given by different severity levels.

Since we use multiple benchmarks for C-OOD detection (i.e., the 11 severity levels), to test the performance models in C-OOD detection, and each severity level may rank the models differently (i.e. the best performers for each severity level may vary), we now consider the question of how these rankings change across severity levels. To this end we calculated the correlations between the rankings obtained at different severity levels. The resulting correlation matrix can be seen in Figure 15. Overall, we observe high correlations, which means that different severity levels generally yield similar rankings of the models. This means that when selecting the best model for deployment, it is usually enough to observe its performance on only a few severity levels.

We also notice that for each severity level $s_i$, the correlation with $s_j$ is higher the closer $j$ is to $i$. This is not surprising and might be anticipated because adjacent severity levels have close severity scores by design.

## G COMPARISON OF DIFFERENT CONFIDENCE FUNCTIONS

This section contains additional technical details and figures related to our comparison of ODIN, max-logit, entropy and MC dropout. Our main conclusions are presented in Section 5 of the main text.

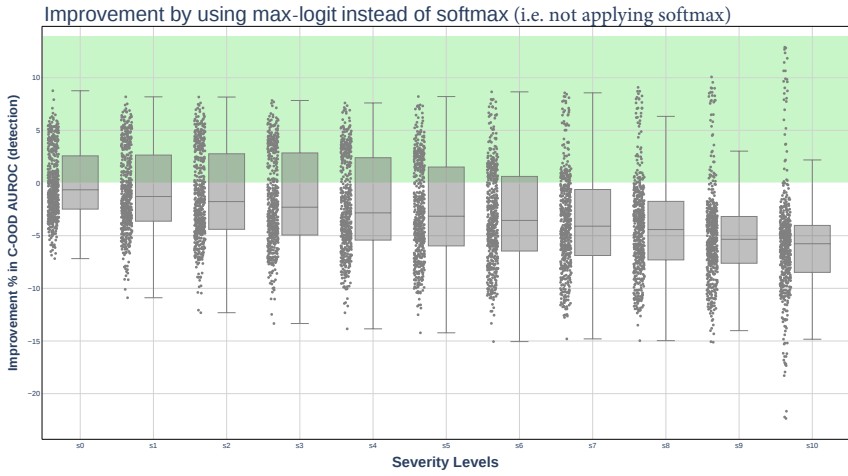

Figure 16: Relative improvement gain in C-OOD detection performance when using max-logit instead of softmax (i.e., not applying softmax). In median terms, using max-logit harms performance over softmax for most evaluated models. However, some models (e.g., ViTs) greatly benefit from not applying softmax. The green shaded area indicates the area of positive improvement.

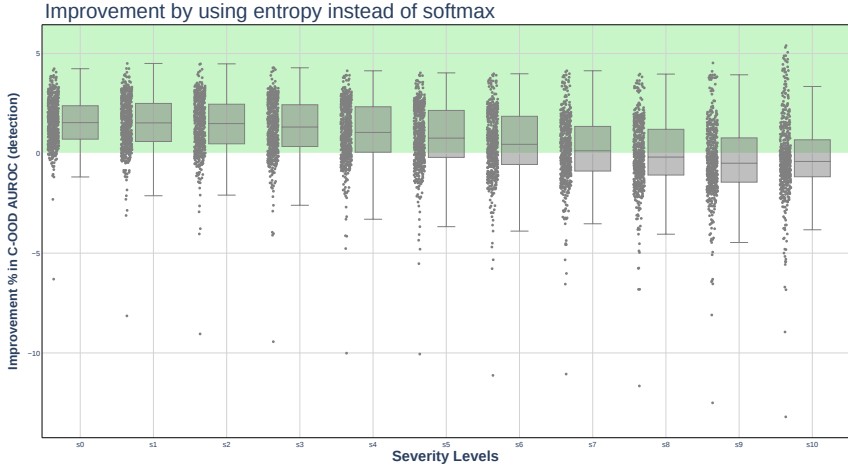

Figure 17: Relative improvement gain in C-OOD detection performance when using entropy instead of softmax. In median terms, entropy offers positive improvement over softmax for most levels of severity except $s \in \{7, 8, 9\}$. The green shaded area indicates the area of positive improvement.

To use MC dropout, we first use 30 dropout-enabled forward passes. The mean softmax score of these passes is calculated and then a predictive entropy score is used as the final uncertainty estimate.

When using ODIN, we use a temperature of 2 and set $\epsilon$ to be $1 \cdot 10^{-5}$. We obtained these hyperparameters by using a simple grid search over a validation set, and using seven models of different

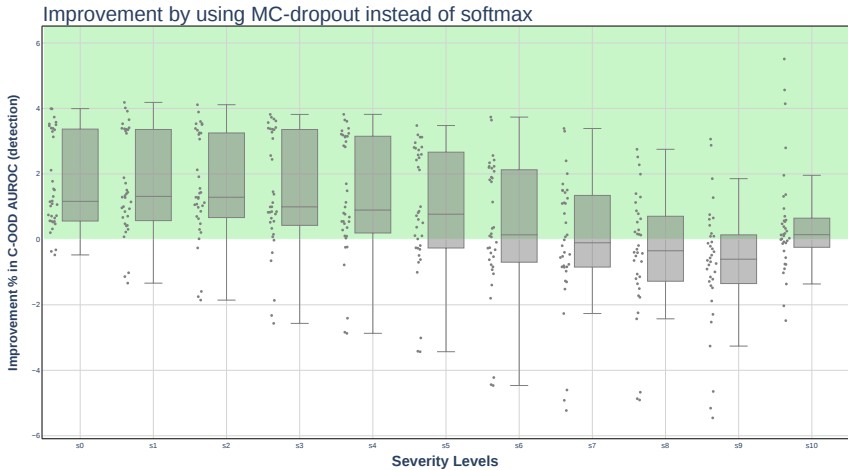

Figure 18: Relative improvement gain in C-OOD detection performance when using MC dropout instead of softmax. We find that MC dropout improves performance, especially at lower levels of severity. The improvement becomes less significant as severity increases.

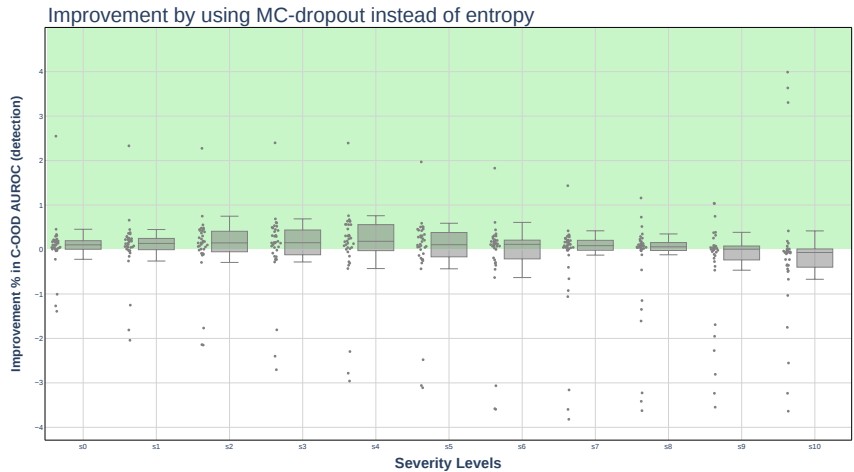

Figure 19: Relative improvement gain in C-OOD detection performance when using MC dropout instead of entropy.

architectures of the entire sample of models evaluated. Our objective was to find the hyperparameters that improve the mean AUROC across all severity levels the most. We believe that fine-tuning the hyperparameters with the specific model and severity levels in mind may allow for better results.

