# OpenReview forum: "A framework for benchmarking Class-out-of-distribution detection and its application to ImageNet"
_ICLR.cc/2023/Conference — ICLR 2023 notable top 25%_

### Official Review · Reviewer_3d28 · 2022-10-24

**Confidence:** 4
**Correctness:** 4
**Technical Novelty And Significance:** 3
**Empirical Novelty And Significance:** 3
**Recommendation:** 8

**Clarity, Quality, Novelty And Reproducibility:**

The quality of this paper is good. The authors present a new benchmarking framework for class-out-of-distribution detection.

**Strength And Weaknesses:**

Strengths:

S1. This work provides a new benchmarking framework to detect class-out-of-distribution instances with different levels of difficulty. This opens new avenues for future OOD detection research.

S2. The authors analyze the results of this benchmarking which lead to numerous interesting observations.

S3. This paper presents a good viewpoint on designing large-scale objective benchmarks with different levels of detection difficulty for class-out-of-distribution, which is useful for real-world applications.

Weaknesses:

W1. It is not clear how and the intuition why a subset of ViTs achieves the best C-OOD detection performance.

W2. The computational complexity is not discussed when analyzing baseline models, e.g., knowledge distillation, and language-vision CLIP.



**Summary Of The Paper:**

This paper proposes a benchmarking framework for class-out-of-distribution detection with various levels of detection difficulty. This work benchmarks this technique's application to ImageNet with 525 publicly available pre-trained ImageNet-1K classifiers. Based on this benchmark, the authors identify several trends in out-of-distribution detection robustness and serve as a foothold for future research.

**Summary Of The Review:**

This is a benchmarking paper focusing on an important problem. The contributions of the paper will have some impact to the OOD detection community.

---

> ### Author Response · Authors · 2022-11-07
> **Authors' response**
>
> Thank you for your positive feedback,
>
> *"It is not clear how and the intuition why a subset of ViTs achieves the best C-OOD detection performance."*
>
> It is a great question, and we hope future research will shed light on this matter.

---

### Official Review · Reviewer_gv9a · 2022-10-24

**Confidence:** 3
**Correctness:** 3
**Technical Novelty And Significance:** 3
**Empirical Novelty And Significance:** 4
**Recommendation:** 8

**Clarity, Quality, Novelty And Reproducibility:**

The paper is high in quality and reproducibility. Then issues with its clarity are fixable and not serious (stated previously in limitations). In terms of the novelty of the paper, while the paper is not the first to address this idea of evaluating models for OOD performance, it does propose a significant advancement in this line of research.

**Strength And Weaknesses:**

The paper well grounded in previous works is attacking a significant problem of interest to the community and has a very extensive empirical validation backing up its claims/observations (which are numerous). The most interesting aspect of the paper for me was the observations and claims about OOD detection, such as the performance of vision transformer models or how knowledge distillation helps with detecting OOD samples. I believe these types of observations are both important for practitioners and also as starting points for additional research.

The paper’s only real weakness is in its clarity, and, possibly its correctness. There are some points in the paper that need further explanation. Specifically

-	In section 2, how is AUC used to measure OOD performance? It seems like there is some cut-off score for $\kappa$ below which the paper assigns the value of OOD versus ID, and then uses these assigned values to compute AUC. If so, what is the cut-off value? Is it different for different $\kappa$’s or models?

-	How does one make apples-to-apples comparisons between models for the OOD performance? As is mentioned in the paper (section 5, point 6) OOD hardness is subjective and the proposed method will have different classes in different hardness ratings for different model + $\kappa$ configurations. This would lead me to believe that one cannot really conclude that one model is better than another in OOD detection as data breakdowns between the two models are not the same. Is there a way to overcome this limitation or otherwise have a comparison between two models on their OOD performance where the only difference is the model itself? Or, is this not a problem for comparing models?

-	A minor point, but could you add the web addresses, probably as footnotes, for timm and torchvision for total clarity on the packages?


**Summary Of The Paper:**

The paper presents a method for benchmarking models (combined with a dataset and a confidence function $\kappa$) for their ability to recognize out-of-distribution examples at test time. In particular, the paper considers the OOD case of when the model is presented with an example with a label that that model had not seen during training. The paper uses the methodology to evaluate 525 different models as well as some different confidence functions to produce some observations about different types of models and about the OOD task generally.

**Summary Of The Review:**

Sometimes the best papers are those papers that raise more questions than they answer. This is one of those papers. While the technical contributions are not especially monumental or flashy, the observations coming from the solid empirical work in the paper are very intriguing, especially from a practicing data scientist lens. As such, I believe this paper does merit acceptance and publication.

---

> ### Author Response · Authors · 2022-11-07
> **Authors' response**
>
> Thank you for your positive feedback,
>
> *"In section 2, how is AUC used to measure OOD performance? It seems like there is some cut-off score for $\kappa$ below which the paper assigns the value of OOD versus ID, and then uses these assigned values to compute AUC. If so, what is the cut-off value? Is it different for different $\kappa$’s or models? "*
>
> We define how AUC is used in the last paragraph of Section 2, formulating it in Equation (1).
> AUC considers all possible cut-off scores, and as such doesn't require assigning a specific one. Additionally, it is also equal to Equation (1) (and could be calculated in that way).
>
> *"How does one make apples-to-apples comparisons between models for the OOD performance? As is mentioned in the paper (section 5, point 6) OOD hardness is subjective and the proposed method will have different classes in different hardness ratings for different model + $\kappa$ configurations. This would lead me to believe that one cannot really conclude that one model is better than another in OOD detection as data breakdowns between the two models are not the same. Is there a way to overcome this limitation or otherwise have a comparison between two models on their OOD performance where the only difference is the model itself? Or, is this not a problem for comparing models?"*
>
> That's a great question.
> In our opinion, our benchmark makes an apples-to-apples comparison from the perspective of hardness, which is more relevant than the specific OOD classes.
> A famous quote says you shouldn't judge a fish by its ability to climb a tree. If we rephrase it to our discussion; if you want to compare a fish with a monkey on what's hardest for them, you should judge the fish by its ability to climb a tree and the monkey's ability to swim (by the way, we know that some monkeys can swim :)    ). Our framework constructs specialized tests for both.
>
> That being said, by considering the construction of severity classes (per model), one can argue that it is possible (neglecting estimation error) to compare the performance of two models specifically for the classes populating their maximal severity (severity 10):
>
> (1) Suppose that model A has better performance (AUROC) on its own group of hardest classes (severity 10), denoted Z, than model B's performance on its own severity 10 classes, denoted Y. Assume that Y does not equal Z (otherwise we are done).
> Thus, AUROC(A,Z) > AUROC(B,Y)
>
> (2) By construction of severity groups, for every set of classes D different than Z, AUROC(A,D) >= AUROC(A,Z). This holds true for any set of classes D, including the set Y. Therefore, AUROC(A,Y) > AUROC(A,Z)
>
>
> By combining (1) and (2) we get that AUROC(A,Y)>=AUROC(A,Z)>AUROC(B,Y) -> AUROC(A,Y)>AUROC(B,Y), meaning that for the same set of classes Y, model A performs better than model B.
>
> A "mirror" argument could be crafted to compare the models' performance on the classes populating their minimal severity (severity 0).
>
> *"A minor point, but could you add the web addresses, probably as footnotes, for timm and torchvision for total clarity on the packages?"*
>
> We added the links to these repositories to the paper (at the end of Section 4).

---

### Official Review · Reviewer_CQ4c · 2022-10-24

**Confidence:** 4
**Correctness:** 3
**Technical Novelty And Significance:** 4
**Empirical Novelty And Significance:** 4
**Recommendation:** 8

**Clarity, Quality, Novelty And Reproducibility:**

- Clarity: Please see weaknesses section. There are some key clarity questions that need to be resolved.
- Quality / novelty: No concerns.
- Reproducibility: Will the authors commit to publicly releasing code for evaluating all models, along with the AUROC per severity level for each model plotted in the paper? This will significantly help reproducibility.


**Strength And Weaknesses:**

Strength:

- The authors point out a key issue in prior work on evaluating OOD detection: when OOD datasets are constructed by filtering to be difficult for one model, the bias in filtering results in other models naturally outperforming the original model.

- The authors evaluate a large array of models on their data, in contrast to prior work, and find interesting conclusions about OOD detection (e.g., that CLIP models outperform models trained on ImageNet)

Weaknesses:

- The notion of severity levels needs to be motivated further. The paper states that the proposed method of computing severity levels is better than prior work – but why do we need severity levels? A plausible alternative metric is to simply compute AUROC on the entire OOD dataset, without the severity levels. Why is this metric insufficient? What conclusions would change if you were to use this metric?
    - If the AUROC overall metric is sufficient, the severity levels can still be used as a diagnostic. It’s unclear that one needs to look at AUROC at each level individually when comparing models.
- While interesting, the result about CLIP’s OOD detection needs further explanation and comparison. The paper claims it outperforms “96%” of other models. But this is a strange metric: the 96% presumably includes models like AlexNet. Using this metric, the ViT-L/32-382 outperforms “99.99%” of other models. It would be more interesting to compare a CLIP model head-to-head with a model with the same architecture, trained with different training strategies, such as training on ImageNet-1k (or pretraining on IN21k then finetuning on IN1K).
- Notation could be more clear – it’s still unclear to me exactly how AUROC is computed.
    - In Eq (1), what is \hat{y}? Is it the prediction of f on x1, or on x2? Is it the same when computing k(x1, \hat{y} | f) as k(x2, \hat{y} | f)? I assume not.
    - In Section 3, k is changed from k(x1, \hat{y} | f) to k(x | f). How is this defined?
    - Is AUROC computed per OOD class, then averaged across classes within a severity level? Or are samples for all classes in a severity level merged together, and then AUROC is computed once for the severity?
- How is ID AUROC defined?
    - Do you compute AUROC per class, with samples from all other classes treated as negative, and then averaged?
- Nits / questions:
    - It would be useful to explain why ODIN, entropy and MC dropout specifically were chosen for evaluation in the work.
    - Fig 6 – what severity level is this evaluated at?
    - Sec 5, (1) – is this on all 525 models?
    - Sec 5, (1) – any ideas why imagenet 21k training hurts OOD detection?
    - Would it be possible to evaluate the effect of removing softmax at test time, and re-evaluating? Depending on how AUROC is defined (see my questions above), this may or may not change the results. This may also allow a more fair comparison between ImageNet models (which usually have a softmax) to CLIP models (which are just using cosine similarity without a softmax). Relatedly, evaluating the impact of the softmax temperature on OOD detection would be interesting. These experiments are not necessary for acceptance, but I would appreciate the authors’ thoughts on the impact of the softmax – it will also help me understand the evaluation better.
    - If possible, it would help to share the AUROC for all models at all severity levels in a table in the supplementary, or in a webpage



**Summary Of The Paper:**

The paper presents a benchmark for out-of-distribution detection for ImageNet(-1K) models, using data from ImageNet-21k as out-of-distribution samples. The work presents a strategy for evaluating OOD detection, avoiding pitfalls encountered in prior work. Additionally, the authors evaluate a wide array of ImageNet models, and highlight findings on which models and confidence functions outperform others.

**Summary Of The Review:**

Overall, this is a good paper, with some issues in presentation and clarity. I have raised some questions about the presentation and clarity above, and particularly about the use of severity levels in the evaluation. Assuming these are addressed, I would vote to accept the work.

---

> ### Author Response · Authors · 2022-11-07
> **Clarifications and reproducibility p1**
>
> Thank you for your detailed and constructive feedback,
>
> *"The notion of severity levels needs to be motivated further. The paper states that the proposed method of computing severity levels is better than prior work – but why do we need severity levels? A plausible alternative metric is to simply compute AUROC on the entire OOD dataset, without the severity levels. Why is this metric insufficient? What conclusions would change if you were to use this metric?"*
>
> We believe the idea of severity (difficulty) levels is important since different tasks and applications may require top performance on particular difficulty levels. A concrete example of this is provided in the last paragraph of Section 3: for a task with a high tolerance for risk (e.g., a task for an entertainment application), the performance of a model on a median severity level might be more important than on the hardest severity level (severity 10).
> We can also imagine that the opposite might be true for some applications with a low tolerance for risk (e.g., medical applications), for which one requires the best performance to be attained even if the OOD is very hard to detect (severity 10). The top performer in terms of AUROC on the entire data might not be the top performer for our desired severity (that is determined by the user's specific task).
>
> Furthermore, in Section 5 we show that detection algorithms do not always improve performance on all inputs equally, and could even hurt performance for specific severity levels and models (see Figure 7 for a striking example). Choosing the combination of (model, detection algorithm) based only on the detection performance on all data may yield sub-optimal results for our specific desired level of difficulty.
>
> We can include in the paper a summary of these arguments to better motivate the need for severity levels.
>
> *"While interesting, the result about CLIP’s OOD detection needs further explanation and comparison. The paper claims it outperforms “96\%” of other models. But this is a strange metric: the 96\% presumably includes models like AlexNet. Using this metric, the ViT-L/32-382 outperforms “99.99\%” of other models. It would be more interesting to compare a CLIP model head-to-head with a model with the same architecture, trained with different training strategies, such as training on ImageNet-1k (or pretraining on IN21k then finetuning on IN1K)."*
>
> Thank you, this is an excellent idea.
> We added a comparative graph (Figure 11) in Appendix B (C-OOD detection with CLIP) of 3 ResNet-50 models: one trained on ImageNet-1k (Torchvision's implementation), CLIP with ResNet-50, and a ResNet-50 trained with a semi-supervised training regime [1].
> Note that we already mentioned in the original version (Appendix B) that in high severity levels CLIP models tend to underperform other models, and so does ResNet-50 trained with CLIP. This new graph also demonstrates the usefulness of severity levels by highlighting the fact that models' performance ranks change across levels.
>
> [1]  Yalniz et al., 2019, Billion-scale semi-supervised learning for image classification
>
> *"In Eq (1), what is \hat{y}? Is it the prediction of f on x1, or on x2? Is it the same when computing k(x1, \hat{y} | f) as k(x2, \hat{y} | f)? I assume not."*
>
> The \hat{y} that appears along with x1 is the prediction for x1, and the one that appears with x2 is the prediction for x2. This is indeed confusing, and we will fix it.
>
>
> *"In Section 3, k is changed from k(x1, \hat{y} | f) to k(x | f). How is this defined?"*
>
> Thanks for finding this. It is defined to be the same. We will make the notation identical in section 3.
>
> *"Is AUROC computed per OOD class, then averaged across classes within a severity level? Or are samples for all classes in a severity level merged together, and then AUROC is computed once for the severity?"*
>
> The latter is correct: AUROC is computed once for each severity.
>
> *"How is ID AUROC defined?
> Do you compute AUROC per class, with samples from all other classes treated as negative, and then averaged?"*
>
> We reduce the multi-class classification into a binary one by treating each prediction as either "correct" (if the predicted label equals the ground-truth label) or "incorrect" otherwise. As it is now a binary classification (of the prediction's correctness), we can easily calculate AUROC.
>
> This is indeed not trivial, and we will include the precise definition for ID AUROC in our revision.

---

> > ### Author Response · Authors · 2022-11-07
> > **Clarifications and reproducibility p2**
> >
> > *"Sec 5, (1) – is this on all 525 models?"
> > *"Sec 5, (1) – any ideas why imagenet 21k training hurts OOD detection?"*
> >
> > As mentioned in Section 5, (1), to make a fair comparison, we only compare pairs of models such that both models have identical architecture
> > and training regimes, with the exception of the method itself.
> > About ImageNet-21k pretraining, that's an interesting question that requires further research.
> > Here is a possible explanation. Some old features (trained over 21K) remain and cause pretrained models to "recognize" the now "unknown" hard classes that are most related to ID classes. Note that severe cases include more related classes by design (see Figure 3).
> >
> > *"Would it be possible to evaluate the effect of removing softmax at test time, and re-evaluating? Depending on how AUROC is defined (see my questions above), this may or may not change the results. This may also allow a more fair comparison between ImageNet models (which usually have a softmax) to CLIP models (which are just using cosine similarity without a softmax). Relatedly, evaluating the impact of the softmax temperature on OOD detection would be interesting. These experiments are not necessary for acceptance, but I would appreciate the authors’ thoughts on the impact of the softmax – it will also help me understand the evaluation better."*
> >
> > Yes, it is possible, and a good idea to evaluate the effect it might have. We will start running this experiment soon and include the results.
> >
> > Using softmax will change the induced partial order between samples' confidence scores.
> > For example, the output logits vector [10, 0.001, 0.001] for a 3-class classification would yield a max softmax score of 0.99 (compared to a max logit score of 10), while the logits vector [1001, 1000, 1000] would yield a max softmax of about 1/3 (compared to a max logit score of 1001). Without using softmax, the order induced by the confidence function would rank the second sample before the first one, and vice-versa when using softmax.
> >
> > While waiting for these results, we note that when we evaluated CLIP when applying softmax to its results (either with or without a "linear-probe", as described in the original paper), its C-OOD detection performance significantly deteriorated. This observation is also mentioned in Appendix B.
> >
> > *"If possible, it would help to share the AUROC for all models at all severity levels in a table in the supplementary, or in a webpage"*
> >
> > Great idea, we will provide this data.
> >
> > *"Reproducibility: Will the authors commit to publicly releasing code for evaluating all models, along with the AUROC per severity level for each model plotted in the paper? This will significantly help reproducibility."*
> >
> > In any case, we plan to release everything publicly upon acceptance. Not only we will provide the results and code to reproduce the benchmarks, but also will share the benchmarks themselves for each of the models we already evaluated (to save computing time for users wishing only to look at or use these benchmarks).

---

> > ### Comment · Reviewer_CQ4c · 2022-11-11
> > **Response**
> >
> > Thanks for your detailed reply, and the follow up experiments. I have a few follow up questions and comments:
> >
> > 1. In Figure 11, what is the overall AUROC of the CLIP trained ResNet-50 vs. the two baseline ResNet-50s? Does the claim still hold that CLIP leads to better OOD detection? If so, I would request that the paper use this evidence when explaining that CLIP achieves good OOD performance. The metric of "better than 96% of other models" is somewhat misleading by contrast. If R50-CLIP does not outperform R50-baselines, then the claim of CLIP being better for OOD should be removed, or at least caveated.
> >
> > 2. I am still unconvinced about the idea of severities as a first-class citizen. I agree severity can be used as a _diagnostic_ to better understand model performance, but why does the top-line metric (AUROC over all the classes) need to care about severity? Wouldn't the evaluation be simpler and more interpretable if the default metric was AUROC overall, and severities were only used as diagnostics?
> >
> > 3. I look forward to the softmax results. If these suggest that CLIP's improvement is due to the lack of a softmax, I request the authors to update the paper to soften the claims of CLIP's OOD ability.

---

> > > ### Author Response · Authors · 2022-11-17
> > > **Softmax, CLIP and severity levels**
> > >
> > > Thank you for your response,
> > >
> > > Although our experiments regarding your idea of examining the effects of not applying softmax are still ongoing, we already have a good number of results to share (over 200 out of 525 models):
> > >
> > > a) In the majority of cases the detection performance changed very slightly (in the range of -2\% to +1\%), with a degradation in performance of more than half of the models.
> > >
> > > b) Only one set of models is significantly improved by not using softmax: this is the ViT subset reported in point 2 of Section 5 to have phenomenal OOD detection performance. These models' AUROC is improved in the range of 6\%-17.6\% (!). This is most notable since these ViTs had already outperformed all other models (CLIPs included) even when they use softmax.
> > >
> > > The fact that softmax could degrade performance in ViTs was interesting for us since, in our experience, softmax is very helpful for ID ranking (ranking incorrect predictions lower than correct ones).
> > >
> > > As for CLIP's relative performance in comparison to other models, not much has changed. The reason is that most of the models slightly deteriorated, with the exception of the subset of ViTs, which have already outperformed CLIP models.
> > > Thus, the relative performance of CLIP models remained almost the same. Nonetheless, we followed your recommendation and have updated the paper and softened the claims about CLIP's performance, focusing on specific observations instead.
> > >
> > > In any case, once we have all softmax results ready we will include them in the paper (we will highlight anything significant, such as the improvement in ViTs, in the main paper and include an in-depth analysis as an appendix). Thank you again for this excellent idea.
> > >
> > > As for AUROC over all OOD classes, the ResNet-50 trained with CLIP has an advantage of above \%1 AUROC over all OOD classes (79.1\% AUROC by CLIP, and 77.8\% and 77.4\% by the baseline and semi-supervised models). We have included this as well.
> > >
> > > As for severity levels. We believe that the model *selection* should be task-specific. For example, if we need a model to classify certain types of butterflies without being confused by very similar butterflies (see Figure 3 for an extreme example, monarch vs. viceroy), we should select the models excelling most at top severity levels. We could also score models by a weighted average of their AUROC across all severity levels, giving more weight to higher severity levels.
> > > A task-specific weighted average could reduce the analysis into one interpretable and task-specific number (by setting the weights). Do you think we should mention this in the paper?
> > > As for using the overall AUROC, following our discussion of CLIP, if we were to select a ResNet-50 model (out of the three we evaluated) only by its overall AUROC, we would select a CLIP although its performance is worse on high severity levels, and would be a sub-optimal choice for our example task.

---

> > > > ### Comment · Reviewer_CQ4c · 2022-11-26
> > > > **Thanks!**
> > > >
> > > > I appreciate the authors' detailed responses. I still disagree with the notion of severity levels as a topline metric -- the topline metric should be AUROC, with severity levels as a diagnostic in my opinion -- but the paper is nevertheless worthy of acceptance, especially given the additional experiments since submission.

---

### Decision · Program_Chairs · 2023-01-20

**Decision:**

Accept: notable-top-25%

**Justification For Why Not Higher Score:**

A spotlight recommendation will help to bring attention to the findings in this work, which is aimed at practitioners in this area.

**Justification For Why Not Lower Score:**

Extensive discussions with the reviewers demonstrate a potential of fruitful discussions at and after the conference, which is great to have. AC agrees with this summary provided by one of the reviewers: 'Sometimes the best papers are those papers that raise more questions than they answer.  And this is one of those papers.'

**Metareview: Summary, Strengths And Weaknesses:**

This paper proses a methodology for evaluating class out-of-distribution detection with various levels of detection difficulty that is model-specific. The main contributions are observations, a.k.a best practices, for OOD benchmarking that are thoroughly validated in an extensive experimental setup. AC agrees with the reviewers that the contributions of the work are important for practitioners and have potential to spark further research in this area. We urge the authors to include the clarifications of the rebuttal into the manuscript, as well as the empirical results on the effects of not applying softmax. Congratulations to the authors!

**Note From Pc:**

if the above contains the word "oral" or "spotlight" please see: "oral" presentation means -> notable-top-5% and "spotlight" means -> notable-top-25%. As stated in our emails, we are disassociating presentation type from AC recommendations